# LED PEDD Discharge Photometry: Effects of Software Driven Measurements for Sensing Applications

**DOI:** 10.3390/s22041526

**Published:** 2022-02-16

**Authors:** Cormac D. Fay, Andrew Nattestad

**Affiliations:** 1SMART Infrastructure Facility, Engineering and Information Sciences, University of Wollongong, Wollongong, NSW 2522, Australia; 2School of Chemistry, Monash University, Clayton, VIC 3800, Australia; andrew.nattestad@monash.edu or; 3Intelligent Polymer Research Institute, AIIM Facility, University of Wollongong, Wollongong, NSW 2522, Australia

**Keywords:** LED, photometry, PEDD, turbidity, timing, discharge, NTU, water, quality, ISO 7027

## Abstract

This work explores the effects of embedded software-driven measurements on a sensory target when using a LED as a photodetector. Water turbidity is used as the sensory target in this study to explore these effects using a practical and important water quality parameter. Impacts on turbidity measurements are examined by adopting the Paired Emitter Detector Diode (PEDD) capacitive discharge technique and comparing common embedded software/firmware implementations. The findings show that the chosen software method can (a) affect the detection performance by up to 67%, (b) result in a variable sampling frequency/period, and (c) lead to an disagreement of the photo capacitance by up to 23%. Optimized code is offered to correct for these issues and its effectiveness is shown through comparative analyses, with the disagreement reduced significantly from 23% to 0.18%. Overall, this work demonstrates that the embedded software is a key and critical factor for PEDD capacitive discharge measurements and must be considered carefully for future measurements in sensor related studies.

## 1. Introduction

LED photometry using the PEDD capacitive discharge technique was originally established as a bi-directional communications method [1] and later explored as a viable colorimetric chemical sensor [2]. This paved the way for other sensing related works such as ammonia [3], iron (II) [4], lead (II) and cadmium (II) [5], pH [6,7], phosphate [8], nitrite [9,10], carbon dioxide [11], oxygen [12], nitrate [13], ethanol and total sulfite [14]. Subsequently, and in line with the above, publications in the bio-sensing field began to emerge. Examples include sweat [15,16,17], hemoglobin [18], human serum [19,20,21], proteins [22,23], glucose [24], dissolved organic substances [25], creatinine in physiological fluids [20], urine [26,27,28], liver function screening [29] and saliva [30].

Throughout this literature, however, there has been no study related to the effects of embedded software implementations [31]. In lieu of this, most reports cite previous studies (including the seminal works of this field) on the subject [1,2]; however, these do not provide extensive or in depth accounts of the implemented software. What is clear is that different use cases have been identified, but not all aspects of this technique have been rigorously established. As timing is a critical aspect for a PEDD discharge measurements, a task the embedded software is responsible for, there are many factors that can impact measurements. This can be significant for the sensing fields requiring quantitative measurements, e.g., the physical/bio/chemical sensing domains.

Given that the bulk of the reported implementations of PEDD in the literature involve sensing [32], it is considered necessary to do the same here in order to explore the effects of software and timing aspects when measuring a sensory target. Turbidity was chosen for this purpose as it is a very practical and important indicator for water quality [32]—both for environmental monitoring [33,34,35,36,37,38,39,40,41,42] and for drinking water [43,44,45,46]. For water bodies such as rivers and lakes in the environment, turbidity is a measurement of the degree of ‘cloudiness’ caused by suspended particles with known contributing sources from human activities such as mining, agriculture or construction [38,39]. These suspended particles can absorb sunlight resulting in an increased water temperature and reduce the amount of light at deeper levels, which negatively impacts aquatic life [35].

It is noted that turbidity involves a direct measurement, without the introduction complicated setups such as filtering and colorimetric reagents for bio/chemical sensing. This simplifies the analysis and reduces the variables, thus allowing this study to focus on the software and timing aspects of the PEDD capacitive discharge measurement technique. Critical factors such as operation implementations (three variations compared)—capable of impacting the timing measurement—will be explored. Additionally, other timing related aspects such as whether the analyte under investigation influences the sampling period, which has otherwise been assumed to be constant, will be fully investigated.

## 2. Materials and Methods

### 2.1. Components and Characterization

The ISO 7027 standard for turbidity measurements outlines conditions under which turbidity should ideally be measured [47], and is used to guide this study. To conform with ISO 7027, a high intensity 860 nm emitter LED with a small emission angle ($\pm 3{}^{\circ }$) was sourced—SFH 4550 (OSRAM Opto Semiconductors, Regensburg, Germany). The LED emission spectrum was determined from the manufacturer’s data sheet, with data extracted using WebPlotDigitizer v4.3 (automeris.io). For the LED in detector mode, its External Quantum Efficiency (EQE) was measured with the device under short circuit conditions using monochromatic illumination (QEX10, PV Measurements, Boulder, CO, USA) from 700 nm to 900 nm, in increments of 10 nm.

A similar process, using the same light source, was performed using the LED in PEDD mode in order to examine and contrast the response from photo-voltaic (PV) measurements. This involved illuminating placing the LED under monochromatic light, again from 700–900 nm in 10 nm increments, but with data collected through the microcontroller rather than the QEX10 DAQ. The attained values were corrected based on the light intensity at each wavelength, measured independently using a calibrated photodiode (Hamamatsu S1010BR, Tokyo, Japan).

### 2.2. System Setup and Design

Positioning of the emitter and detector LEDs were controlled using a chamber designed in CAD (FreeCAD, 0.18.4, Juergen Riegel, Werner Mayer, Yorik van Havre, http://www.freecadweb.org, accessed on 3 February 2022) as seen in Figure 1, which was fabricated using a 3D printer (Flashforge Dreamer, Zhejiang Flashforge 3D technology Co., Ltd., Jinhua, China). The chamber was designed to hold a standard 10 mm cuvette, similar to the physical arrangements found in standard bench top spectra-photometers. The LED electrical schematic is presented (KiCAD, 5.1.7, Jean-Pierre Charras, http://www.kicad.org, accessed on 3 February 2022) illustrating the connections to the micro-controller board (Arduino Nano v3.0, Atmega328, purchased at Jaycar Electronics, http://www.jaycar.com.au, accessed on 3 February 2022).

### 2.3. LED Detection Principle

The principle of detecting luminous intensity via a LED in reverse bias using a microcontroller has been known for some time now [1,2]. For context, Figure 1 shows the circuit schematic of the detector LED and Figure 2 presents an annotated version of a typical charge/discharge profile. The file ‘firmware.ino’, available in the Appendix A, provides the embedded code used in this study, with Table 1 listing all of the relevant functions associated with the undertaken experiments. The code was developed from an exact description of the process reported in the literature [1,2,48,49] and will be denoted as the ‘uncorrected $T_{s}$’ approach for reasons that will become evident later. Briefly, the detector LED is first charged in reverse bias mode (Figure 1) by the microcontroller’s I/O being set as an output for 100 ms. This time was heuristically determined for the LED used in this study. Subsequently, the I/O is switched to input mode, allowing the photo-capacitance to discharge over time ($t_{Discharge}$), see Figure 2. As it discharges, the logic state of the I/O is repeatedly checked ($t_{C_{0}}\ldots t_{C_{N}}$) and a software counter (typically 16-bit) is incremented when above the logic threshold. For example, in the case provided in Figure 2, the counter would report 7 as the measurement value. In addition to this, the total time taken for a measurement to take place (sampling period, $T_{s}$) is also recorded to adhere to the format of results expressed by studies in the literature [48,50].

### 2.4. Turbidity Measurement

The 1000 NTU and 100 NTU turbidity calibration standards were sourced (Formazin, TURBP1000 and TURBP100, respectively) from Sigma Aldrich, stored in a refrigerator until being diluted using purified water (Milli Q) to produce varying turbidity concentrations. The 0–1000 NTU range was prepared in steps of 200 NTU and within this a 0–100 NTU range was prepared in steps of 20 NTU to examine a lower sensing range. The solutions were transferred into fresh cuvettes where the maximum (1000 NTU) and minimum (0 NTU) suspension cuvettes were placed in the fabricated chamber (Figure 1). Here, a resistor of 56 k Ω in series with the emitter LED was found to maximize the dynamic measurement range of the detector. After that, each suspension was measured in turn and ∼10 measurements were recorded. This process was repeated in triplicate in order to investigate reproducibility.

### 2.5. Embedded Software Implementations

Table 2 presents three methods used to read the I/O status and perform the comparison check during a measurement. As mentioned, the embedded software used in this study, with the identified function listed in Table 1 can be found in the ESI, all of which follow the discharge model discussed in Section 2.3. For reading the I/O status, Method 1 uses the Arduino standard library function ‘digitalRead()’, while Methods 2 and 3 accesses the registry directly using bitwise operators. For the comparative operation, the ‘if’ statement is the most commonly used and is part of Methods 1 and 2, while the switch statement was considered and implemented as part of Method 3. While it can be argued that the Wiring style can implement the switch statement, any effect will appear as a difference between Methods 2 and 3.

## 3. Results

### 3.1. LED Spectral Sensitivity

Figure 3 presents the emission spectra (at I*_F_* = 100 mA) of the SFH 4550 LED, along with its electrical response, measured in PV mode and in PEDD mode. A 20–30 nm Stokes shift is seen between the emission peak and the peak of the EQE in PV mode, which is typical for inorganic semiconductors (see Figure A1 in Appendix A for the JV Curve), and similar to observations by Anh-Bui [51], Li [52] and Tymecki [49,53]. It is noted that the EQE spectral band (PV mode) is wider than the emission by ∼60–70 nm, which is counter to claims in the seminal study of Lau et al. that “*an LED is sensitive to all wavelengths of light equal to or shorter than the emission wavelength*” [2]. On the other hand, such a response agrees with other reports [51,53], which clearly showed that the responses from LEDs in PV detector mode are poor at lower wavelengths, and typically have a FWHM of only ∼40–60 nm. While it is known that most semiconductors have very broad absorption for photons of greater than the bandgap energy, this may be mitigated by other components in the LEDs, such as charge selective layers, which may absorb some of these wavelengths before the photons can reach the active material. It is noted that in some of Lau’s subsequent work [54,55], a multi-wavelength emitter LED system was used, with a single IR detector and a sensitivity of greater than ∼60–80 nm.

The PEDD mode measurement is shown here as the reciprocal of the time constant for each wavelength, divided by the intensity determined by our calibration diode. The rationale for this approach is explored in further detail in the ESI, see Figure A2, Figure A3 and Figure A4 in Appendix B for this process. There is considerable disagreement between the modes of measurement, i.e., PEDD discharge and PV modes, with, to the best of our knowledge, no previous studies comparing these different modes. This difference may be, at least in part, due to non-linear responses (with respect to light intensity) in PV mode, which is explored in more detail in the ESI.

This could also be due to the use of a reciprocal value increasing sensitivity to the absorption tail. Consider that the PV measurements relies upon light induced generation/recombination and exceed the energy threshold in order to surpass the band gap, while in the PEDD mode, there may be more opportunity for trapped charges (related to defect states) to facilitate relaxation. While outside the scope of this study, this highlights that further, in depth, investigations into LED spectral range/sensitivities (including differences in measurements being conducted via PV and PEDD mode) is warranted. Ultimately, for this study, the primary purpose of Figure 3 is to validate that the emitter and detector (in PEDD mode) are within the ISO recommended spectral range [47] and that they overlap—perhaps more so in PEDD than PV mode.

### 3.2. Turbidity Measurements

#### 3.2.1. Literature Derived Method (Uncorrected $T_{s}$)

Figure 4a,b presents the calibration of the turbidity samples using the procedures derived from the literature, outlined earlier in Section 2.3. The response is expressed in Figure 4a in terms of discharge counts, in line with the bulk of studies in the literature [10,11,12,56]. Each of these provide a response that can be well fitted to a power growth function using a model of $y=Ax^{\tau }+y_{0}$ ($R^{2}\geq 0.998$), with parameters reported in Table 3. The data show that each method varies in their discharge count quantity when examining the same turbidity concentrations. This can be ranked from highest to lowest as Method 3, to 2, to 1, which implies that Method 3 can perform operations faster than Methods 2 or 1 and therefore can offer a higher resolution. As a result, if one wants to achieve the highest resolution and/or perform multiple measurements with a minimized sampling period then the ‘switch-memoryAddressing’ operation is recommended.

Considering that timing is the key factor in estimating the turbidity concentration using a PEDD setup, measurement of the samples was repeated and the sampling periods (duration for full measurements) were recorded, see Figure 4b and Table 3 for model parameters. While there is a difference in terms of implemented method, the sampling period was not constant and in fact varied for samples with different turbidities. This will introduce error as the PEDD measurement technique is time based on the discharge time, inferred from the number of cycles counted. The significance of this is that the response is not proportional to the turbidity concentration alone, but rather to an additional influencing factor attributed to the operational timing, which appears to be exponential in nature. When performing successive measurements, the very fact that the sampling period can vary, temporal data processing such as Fourier analysis/filtering would therefore not be possible.

#### 3.2.2. Proposed Method (Corrected $T_{s}$)

The most likely reason for the timing issues observed previously is based on the software operations. Referring back to Figure 2 for explanatory purposes, prior to the discharge profile crossing the logic threshold the operation includes the software increment, as described in Section 2.3. While such operations are rapidly executed, it is not clear whether this could have an effect on determining the sampling period. In the case of Figure 2, there would be a larger temporal weighting on the first 8 I/O logic checks ($t_{C_{0}}\ldots t_{C_{7}}$), and less thereafter ($t_{C_{7}}\ldots t_{C_{N}}$). In order to investigate this, an increment operation was added when the discharge profile was below the logic threshold (see ESI and Table 1 for the relevant code) with the experiments repeated.

Figure 4c,d presents the calibration of the system with respect to the turbidity standards, which was implemented here in order to achieve a constant sampling period. As before, power trends are suggested in Figure 4c and confirmed via fitting of the model, $R^{2}\geq 0.998$ (see Table 3). Please note that methods 2 is difficult to see in Figure 4c. The reason for this is because methods 2 and 3 are in very good agreement (as per parameters in Table 3), resulting in method 3 obscuring method 2’s presentation. For the discharge count data, it appears that there are less equivalent discharge counts (vertical axis) than shown in Figure 4a, which implies a lower sampling rate and therefore a lower resolution—also indicated through the parameters in Table 3. This directly confirms that the time for an increment to take place does affect the measurement. What is lost in resolution, however, is gained through a constant sampling frequency as seen in the parameters and Figure 4d, with a range visually less than the timing data in Figure 4b. The proposed correction allows for a minimization of the sampling period variation, which was observed from the previous approach.

#### 3.2.3. Quantitative Comparison

Figure 5 presents a comparison of the turbidity measurements between the literature derived (uncorrected $T_{s}$) approach (Figure 4a,b) and the proposed (corrected $T_{s}$) approach (Figure 4c,d). Figure 5a shows that there is a clear difference arising from the software method employed, which is in line with observations discussed previously. For uncorrected turbidity measurements, the resolution was affected by ∼67% (method 1) and 26% (method 2), with respect to method 3. When employing the proposed approach, the relative discharge counter difference (with respect to the uncorrected Method 3 data) decreased to 30% (method 1) and 63% (methods 2 and 3), demonstrating that the uncorrected approach leads to softwares being executed differently, with relative performance extending to 70%. For the timing of the measurements, Figure 5b shows that the sampling period range ($T_{s_{max}}-T_{s_{min}}$) differs with the uncorrected approach (as much as 67%), while with the correction in place the sampling period has less than 1 μ s difference over the entire range. This is a strong indication that the sampling period varies considerably with respect to the uncorrected approach by demonstrating that the embedded programming operations can affect the timing considerably, which has not been addressed by the sensor literature. This is supported by the data from the corrected approach where the discharge timing sampling period difference is considerably and relatively small.

#### 3.2.4. Comparison via Photo-Capacitance

In order to further explore the timing aspects as discussed above in more detail, the photo-capacitance of each measurement per method was examined for both the literature derived approach (Figure 6a) and the proposed approach (Figure 6b). This was calculated by the fitted model: $V=V_{0}e^{-t/\tau }$ ($\tau =T_{NTU}C$, $V_{0}=5\;V$, $V_{TH}$ = 1.5V), with the photo-capacitance expressed as $C={-}^{t}/T_{NTU}\cdot \ln\left(1.5/5\right)$.

For the literature derived/uncorrected $T_{s}$ approach in Figure 6a, it can be seen that there is a disagreement between the photo-capacitance calculated using each method, which is more prominent at lower NTU values, as shown in the inset. Considering that the physical processes and parameters are constant at each turbidity concentration, the calculated photo-capacitance should agree regardless of the method employed. In this case the methods disagree by up to 23% or 160 pF, which is a factor that can impact estimations of the sensory target.

For the proposed/corrected $T_{s}$ approach in Figure 6b, the results were processed in the same way as before—to calculate the photo-capacitance. This shows that regardless of the implemented method, the estimated photo-capacitances are in excellent agreement, with a disagreement as low as 0.18% when compared to 23% in the uncorrected $T_{s}$ approach. Given the strong agreement between each methods, due to the corrected approach, a calculated percentage relative error between both approaches was found to be 9%, 20% and 26% for Methods 1, 2 and 3, respectively.

#### 3.2.5. Graphical Comparison

In order to understand the implications of the timing effects more clearly in a graphical manner, consider that when measuring the same turbidity concentration the LED discharge profile should remain be identical, regardless of the software (method or correction) approach adopted. Figure 7 (middle) presents the measured values and extrapolated discharge profile of the 0 NTU and 1000 NTU samples expressed in the time domain. The corresponding results from the uncorrected approach is shown in Figure 7 (bottom) as horizontal bars. For each method a pair of bars of the same color (conforming with the color convention adopted previously and identification number as per Table 2) represent the results from the 0 NTU sample (top bar) and the 1000 NTU sample (bottom bar). Each bar’s full width represents the sampling period, while the solid color (left segment) represents the proportion above the logic threshold, in line with the vertical dashed lines provided in the discharge profile. From this, it is clear that the reported time varies with respect to turbidity concentration (pronounced colored section), which is expected. Additionally, the sampling period (full bar width) also varies with respect to turbidity concentration. This shows that with the uncorrected $T_{s}$ approach, the signal is not proportional to the turbidity concentration alone. For the corrected $T_{s}$ approach, it is clear that there is strong agreements between the 0 and 1000 NTU sampling periods independent of the implemented method, which means that the resulting signal is directly proportional to the turbidity concentration under investigation and has removed the second/unwanted proportionality issue, albeit at the expense of a slightly extended $T_{s}$.

The implications of this allows for temporal data processing such as Fourier analysis/filtering, which would not be possible with the uncorrected $T_{s}$ approach. This can be a significant issue when performing successive measurements if there is a rapid change in conditions. While the implementation of a hardware timer can control the starting time of the discharge and counter/timer, it cannot address the secondary issue—that the sampling period varies with respect to turbidity concentration, yet the proposed correction proposed herein evidently has.

One aspect of PEDD research that has gone unaddressed in the literature is the inability to assess implemented methods due to unpublished code. When following the methodology section exactly, it is clear that there are differences in implementable methods and a constant/non-constant sampling period approach, which can evidently affect the results as shown previously in this study. An interesting question arising from this is reproducibility of the studies given the unknowns and relative differences between methods and approaches. To investigate this possibility in the context of turbidity estimations Figure 8 presents a heat map comparing all methods, i.e., each of the three methods and both uncorrected and corrected sampling periods. To explain, the fitted model of one approach/method was used to predict the turbidity concentration using the measured values of another. The presented values represent the maximum percentage relative error of this prediction. When each method is compared against itself, the values are 2.2%, which is relatively low. Depending on the approach/method, this can extend up to 45% error in the case of using uncorrected method 3’s model to estimate the corrected method 1’s turbidity data. This opens up the question as to the reproducibility of previously published studies in the literature given the incomplete or unpublished code.

### 3.3. Discussion

It is understandable why studies in the literature have not accounted for the timing issues discovered in this study. In many cases, the primary focus of reported studies has been to explore the capabilities of the PEDD approach when investigating another sensory target/application. In sensing studies a calibration model is achieved with the adoption of the general PEDD detection procedure [1,2,48], although the embedded software is not reported. This is concerning as without a complete account one cannot reproduce these studies, as different software implementations can impact sensor characteristics such as resolution, LOD or sensitivity, which therefore cannot be fully verified. For instance, Table 4 presents these sensory characteristics for each approach and method explored in this study [57]. In the discharge time domain it can be seen that the corrected $T_{s}$ approach yields more appealing characteristics with higher sensitivities, lower limits of detection and a greater agreeable range compared to the uncorrected $T_{s}$ approach, which is in line with previous discussions. This also shows that there is a larger disagreements between the uncorrected methods, which only illustrates the need for a standard method of measurement for PEDD sensing. Furthermore, note that better analysis appears to take place using the discharge time domain over the counter domain. While the counter results shown in Figure 4a,c appear to show less sensitivity, when converted to an SI measurement unit (ms in this case) a more reliable analysis can result. It is therefore recommended to adopt the time domain for future analyses.

As the work in this study has shown, there is no ubiquitous procedure when it comes to PEDD implementations with the choice of methodology and timing demonstrated herein to have a considerable effect on the response characteristics. The PEDD capacitive discharge technique has been identified as a low-cost and cost-effective method for sensing purposes [58] and allowed for a large number of studies to take place, as reviewed in the introduction section. The embedded software has, however, been a missing reported factor and demonstrated herein to play a significant role. Given that the sensing domain has focused primarily on chemical and biological sensing, it would be interesting to explore the software effects on targets such as those highlighted earlier in the introduction or by others in various reviews on the subject [50,59,60,61].

The timing effects would also be important to the research community adopting LEDs as optical communication devices [62,63,64] and warrants exploration. In all applications outside of controlled environments, noise can be a considerable issue for any detection device. While a detector LED can be quite sensitive [8,51], it relies on a very small current—often discharged from a capacitance in the pico-Farad range—and therefore prone to electromagnetic interferences, which may be one reason for large error bars found in previous studies [5,65]. Filtering this is not possible with a variable sampling period, which highlights the importance of a constant sampling period/frequency and therefore highlights the importance of the work in this study. Finally, the full exploration of all factors associated with the PEDD capacitive discharge technique is necessary for this to become a sensing standard.

## 4. Conclusions

This work has demonstrated the impact of the embedded software in measurements of a sensory target when implementing the PEDD capacitive discharge technique. This was done by first performing absorbance measurements of turbidity samples from 0–1000 NTU, based on the operation described in the literature, and implementing multiple software methods to achieve this. The choice of method involved was shown to affect the detection performance in terms of resolution by up to 67%, which can therefore affect the LOD and sensitivity detection capabilities. The results have also shown that the sampling period was variable with respect to analyte concentration. This was demonstrated by a disagreement of the photo-capacitance by the adopted methods of up to 23%. Timing effects were recognized and corrected using the proposed approach in this study, which resulted in a constant sampling period/frequency, evident from a maximum disagreement between the photo-capacitance estimations of less than 0.185%. A comparison between both approaches has shown that the error associated with the uncorrected approach can be as high as 26% (%RE), meaning that the implemented software, and operations thereof, play a significant role in the accuracy of PEDD capacitive discharge measurements and must be taken into consideration if a standard is to be accepted by the sensing community—or indeed established by standards organizations.

## Figures and Tables

**Figure 1 sensors-22-01526-f001:**
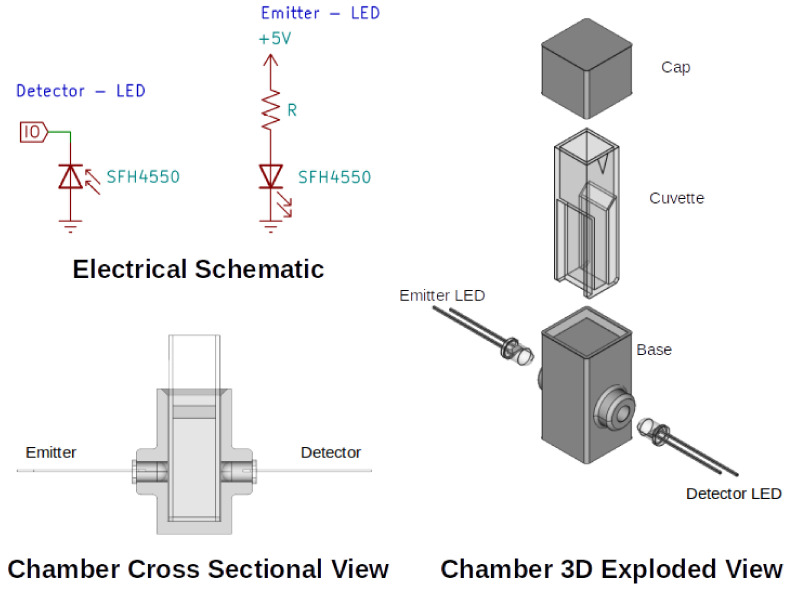
CAD drawings illustrating the electrical schematic/connections and mechanical arrangement of the LEDs, cuvette and 3D printed cuvette holder.

**Figure 2 sensors-22-01526-f002:**
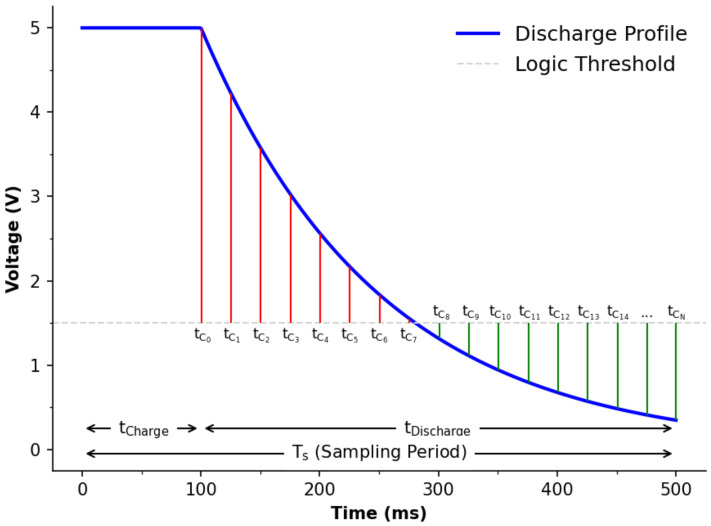
LED voltage model showing a charge and discharge profile for explanatory purposes.

**Figure 3 sensors-22-01526-f003:**
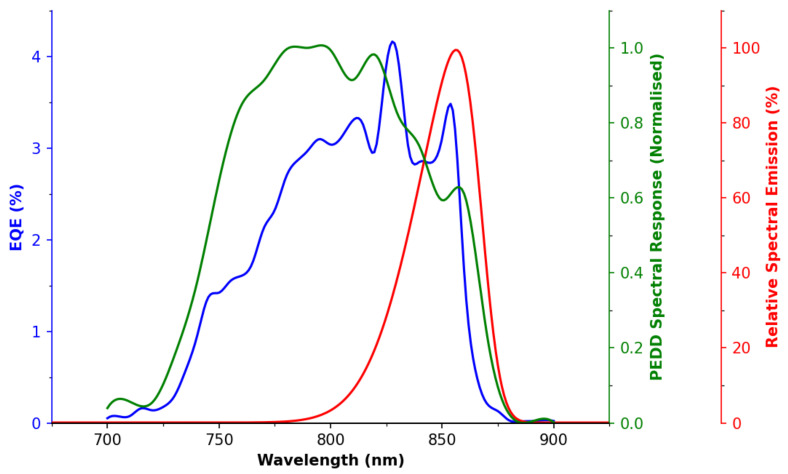
Spectral sensitivity and emission of the LED from 700–900 nm. EQE response (blue line), PEDD in measurement mode (green line) and LED emission spectra (red line).

**Figure 4 sensors-22-01526-f004:**
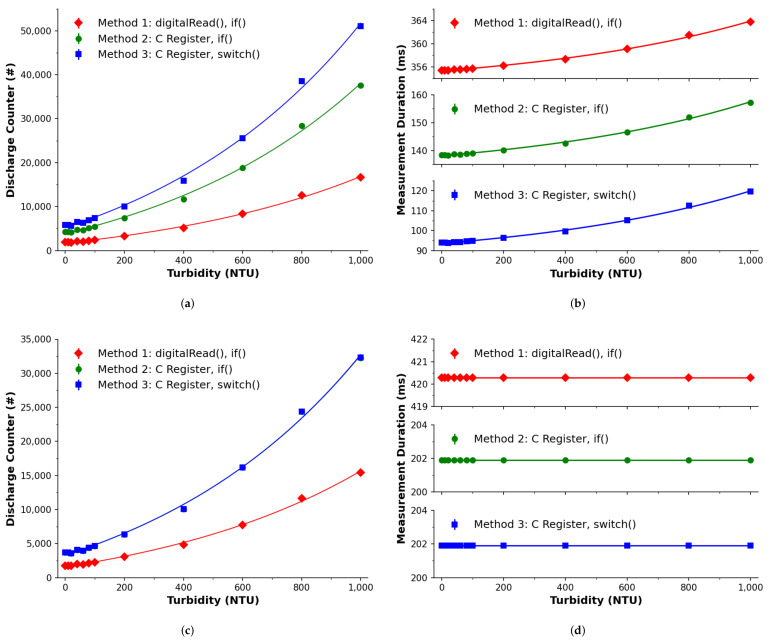
Turbidity calibration of the PEDD system adopting the literature derived (top: **a**,**b**) and proposed (bottom: **c**,**d**) approaches and compared using three software implementation methods. The average of multiple measurements are represented by the markers the standard deviation is represented by the error bars, and the lines show a fitted power model ($y=Ax^{\tau }+y_{0}$). Method 1 (⧫), Method 2 (●) and Method 3 (■) are implementations of the parameters listed previously in Table 2. (**a**) Software counter: uncorrected $T_{s}$. (**b**) Measurement duration: uncorrected $T_{s}$. (**c**) Software counter: corrected $T_{s}$. (**d**) Measurement duration: corrected $T_{s}$.

**Figure 5 sensors-22-01526-f005:**
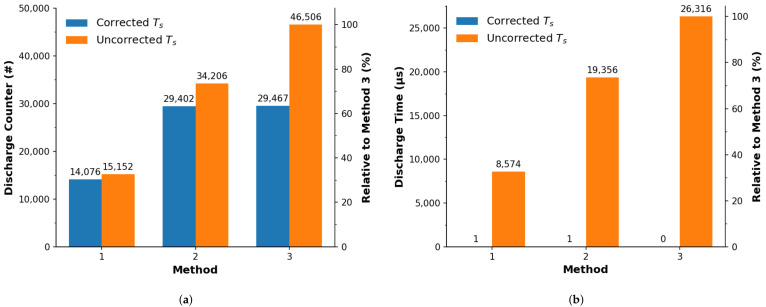
Comparison between the uncorrected and corrected $T_{s}$ approaches calculated from the data sets shown in Figure 4. (**a**) Discharge counter. (**b**) Discharge time.

**Figure 6 sensors-22-01526-f006:**
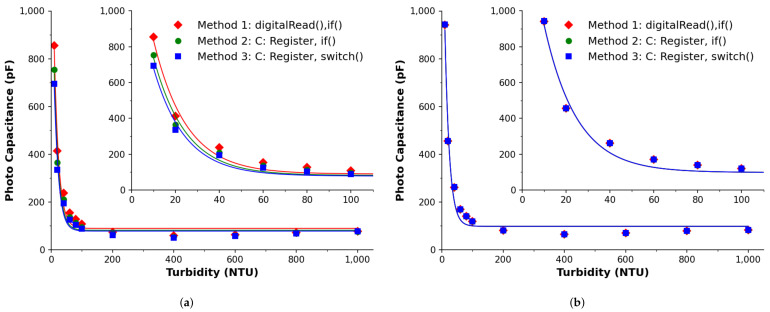
Photo-capacitance discharge profiles derived for each applied method for: the literature derived approach (**a**) and proposed approach (**b**). Method 1 (⧫), Method 2 (●) and Method 3 (■) are implementations of the parameters listed previously in Table 2. (**a**) Photo-capacitance: uncorrected $T_{s}$. (**b**) Photo-capacitance: corrected $T_{s}$.

**Figure 7 sensors-22-01526-f007:**
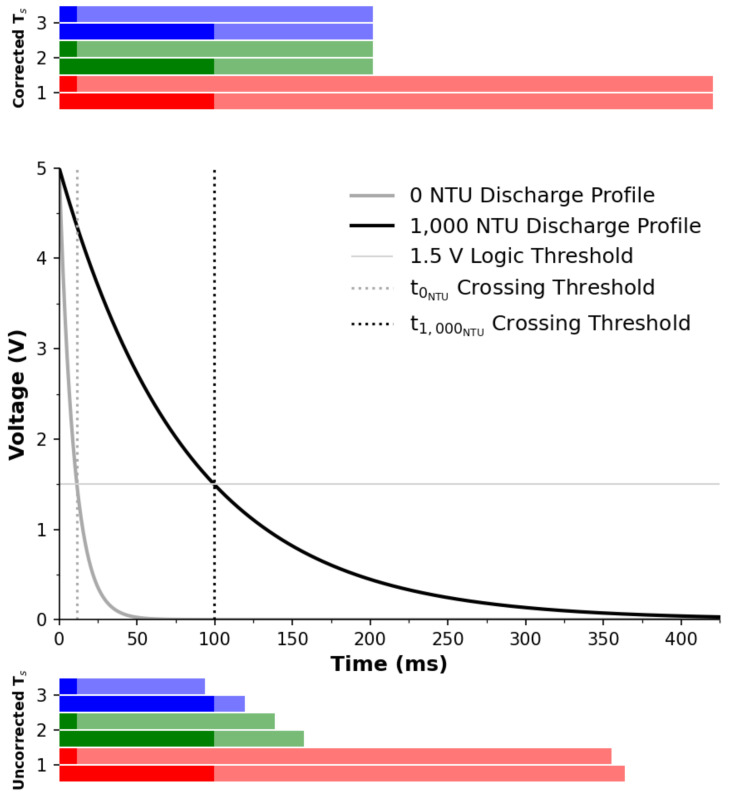
Time domain view of the 0 and 1000 NTU measurements. The full width of the bars show the sampling period. The pronounced color area of the bars indicates the turbidity concentration. **Top**: Corrected sampling period approach. **Middle**: Extrapolated discharge profiles with logic and crossing thresholds. **Bottom**: Uncorrected sampling period approach. Colour key: Method 1 (red), Method 2 (green), Method 3 (blue).

**Figure 8 sensors-22-01526-f008:**
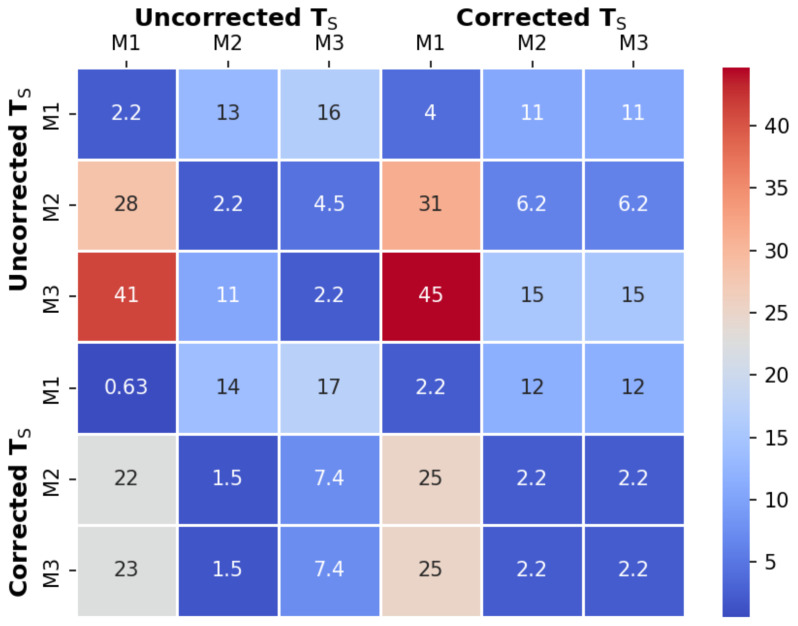
The relative error between methods and approaches when estimating the turbidity concentration using the fitted models from the data in Figure 4a,c and Table 3.

**Table 1 sensors-22-01526-t001:** Lookup table for identifying the appropriate function in the ESI ‘firmware.ino’ file.

$T_{s}$	Measurement	Method	Function
Uncorrected	Counter	1	uncorrectedCounterMethod1()
		2	uncorrectedCounterMethod2()
		3	uncorrectedCounterMethod3()
	Timer	1	uncorrectedTimerMethod1()
		2	uncorrectedTimerMethod2()
		3	uncorrectedTimerMethod3()
Corrected	Counter	1	correctedCounterMethod1()
		2	correctedCounterMethod2()
		3	correctedCounterMethod3()
	Timer	1	correctedTimerMethod1()
		2	correctedTimerMethod2()
		3	correctedTimerMethod3()

**Table 2 sensors-22-01526-t002:** List of three methods for PEDD software implementation.

	Method		
	1	2	3
Style	Wiring	C Bitwise	C Bitwise
Operation	digitalRead()	PIND & PD2	PIND & PD2
Statement	if	if	switch

**Table 3 sensors-22-01526-t003:** List of model parameters for the data presented in Figure 4 for the uncorrected and corrected measurement sampling period $\left(T_{s}\right)$.

$T_{s}$	Coeff.	Counter			Timer		
		1	2	3	1	2	3
Uncorrected	*A*	5504	12,568	16,906	3118	7115	9566
	τ	756	761	756	757	761	756
	$y_{0}$	−3810	−8733	−11,706	352,200	131,036	84,035
	$R^{2}$	0.998	0.998	0.998	0.998	0.998	0.998
Corrected	*A*	5144	10,721	10,671	0	0	0
	τ	759	758	755	1	1	1
	$y_{0}$	−3566	−7433	−7382	420,286	201,908	201,907
	$R^{2}$	0.998	0.998	0.998	0.327	0.039	0.006

**Table 4 sensors-22-01526-t004:** List of calculated sensory characteristics: sensitivities, limits of detection (LOD) and range for each method and approach.

T_s_ Approach	Characteristic	Method		
		1	2	3
Uncorrected	Sensitivity (μ s/NTU)	40.51	42.13	46.01
	LOD (NTU)	1.61	1.17	0.46
	Range (ms)	85	82.3	81.3
Corrected	Sensitivity (μ s/NTU)	50.26	50.37	50.41
	LOD (NTU)	0.17	0.14	0.12
	Range (ms)	88.1	88.4	88.6

## Data Availability

Not applicable.

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
