# Peer review of "LED PEDD Discharge Photometry: Effects of Software Driven Measurements for Sensing Applications"

_sensors, 2022, doi:10.3390/s22041526_

Round 1

Reviewer 1 Report

This manuscript had demonstrated the impact of embedded software on the PEDD capacitive discharge technique in measurements such as turbidity. As the result, it was shown the resolution was improved up to 67%.  However, the detection range, limit of detection (LOD) and sensitivity are the mostly concerned parameters for sensing. I suggest these parameters should be given in the example of turbidity measurement to verify the conclusion of this work.

Reviewer 2 Report

The authors have illustrated the effects of software-driven measurements of water turbidity while using LED as a detector. The work is interesting, and the manuscript is well written. I have some minor comments as:

Can the abbreviations list be pushed up right after abstract?.

Include the term water turbidity in the abstract. As that is the system under concern, it is important to include it in the abstract.

Try to add one line regarding the commonly known impurities which contribute to the turbidity of water.

Increase the font size of the axis labels.

Try to include a few unknown samples in addition to the known sample to show the efficiency of the proposed method.

There is no mention of the supplementary data in the main text.

If the authors take care of these minor comments then I recommend the manuscript for publication.

Round 2

Reviewer 1 Report

The authors provided a satisfied response to the comment. Sensitivity, limit of detection and detection range were given in the revised manuscript for each method and the results  made sense.